# A Reduced Tryptophan Diet in Patients with Diarrhoea-Predominant Irritable Bowel Syndrome Improves Their Abdominal Symptoms and Their Quality of Life through Reduction of Serotonin Levels and Its Urinary Metabolites

**DOI:** 10.3390/ijms232315314

**Published:** 2022-12-05

**Authors:** Cezary Chojnacki, Marta Medrek-Socha, Aleksandra Blonska, Radoslaw Zajdel, Jan Chojnacki, Tomasz Poplawski

**Affiliations:** 1Department of Clinical Nutrition and Gastroenterological Diagnostics, Medical University of Lodz, 90-647 Lodz, Poland; 2Department of Computer Science in Economics, University of Lodz, 90-237 Lodz, Poland; 3Department of Pharmaceutical Microbiology and Biochemistry, Medical University of Lodz, 90-136 Lodz, Poland

**Keywords:** irritable bowel syndrome, tryptophan intake, serotonin, melatonin

## Abstract

(1). An essential component of any treatment for patients with irritable bowel syndrome (IBS) is an adequate diet. Currently, a low FODMAP diet is recommended as a first-line therapy, but it does not relieve abdominal discomfort in all patients, and alternative nutritional treatment is required. The purpose of this study was to evaluate the effect of a tryptophan-lowering diet (TRP) on abdominal and mental symptoms in patients with irritable bowel syndrome with predominant diarrhea (IBS-D). (2). The study included 40 patients with IBS-D, and 40 healthy subjects served as a baseline for IBS-D patients, after excluding comorbidities. The TRP intake was calculated using the nutritional calculator. The severity of abdominal symptoms was assessed using the gastrointestinal symptom rating scale (GSRS-IBS). Mental state was assessed using the Hamilton anxiety rating scale (HAM-A), the Hamilton depression rating scale (HAM-D), and the insomnia severity index (ISI). The serum levels of serotonin and melatonin and the urinary excretion of their metabolites 5-hydroxyindoleacetic acid (5-HIAA) and 6-sulfatoxymelatonin (aMT6) were determined by the ELISA method. The severity of symptoms and laboratory data were analyzed before and after a 12 week diet with tryptophan restricted to a daily dose 10 mg per kilogram body weight. (3). Compared to the control group, patients with IBS-D had a higher serum level of serotonin (198.2 ± 38.1 vs. 142.3 ± 36.4 ng/mL; *p* < 0.001) but a similar level of melatonin (8.6 ± 1.1 vs. 9.4 ± 3.0 pg/mL; *p* > 0.05). The urinary excretion of 5-HIAA was also higher in patients with IBS-D patients (7.7 ± 1.5 vs. 6.0 ± 1.7 mg/24 h; *p* < 0.001). After nutritional treatment, both the serum serotonin level and the urinary 5-HIAA excretion significantly decreased (*p* < 0.001). The severity of the abdominal symptoms and anxiety also decreased, while the HAM-D score and the ISI score remained unchanged (4). Lowering the dietary intake of tryptophan may reduce abdominal complaints and does not alter the mental state of IBS-D patients.

## 1. Introduction

Irritable bowel syndrome (IBS) is a functional disease characterized by recurrent abdominal pain associated with changes in bowel habits. Several pathogenic factors have been postulated for the occurrence of IBS, including gastrointestinal dysmotility, visceral hypersensitivity, the gut–brain axis, changes in the gut microbiota, immune activation, and others [1]. These different factors influence the clinical picture of this syndrome. Four subtypes of IBS are distinguished, namely, predominant constipation (IBS-C), predominant diarrhea (IBS-D), mixed intestinal habits (IMS-M), and unclassified (IBS-U), according to Rome IV criteria [2]. The implication of various types of the disease is a number of different treatment options. Treatment consists of behavioral modification, psychotherapy, non-pharmacological interventions, pharmacotherapy, and other methods [3,4]. An elementary element of any treatment is nutritional intervention [5,6,7]. A gluten-free diet is often recommended in patients with IBS, but treatment results are satisfactory only in some patients [8]. It is not clear whether gluten or the other components of wheat determine the symptoms of IBS. A very-low-carbohydrate diet (VLCD) also improves symptoms, such as abdominal pain and bowel movement, in patients with IBS-D [9]. Another diet excludes lactose (LFD), but the results of current studies suggest that IBS symptoms were independent of lactose maldigestion [10,11]. Similarly, the fructose-free diet (FFD) relieves abdominal symptoms only in some patients with IBS-D [12]. In recent years, a low FODMAP (fermentable oligosaccharides, disaccharides, monosaccharides, and polyols) diet has been recommended for IBS-D [13,14]. This diet provides a reduction in the consumption of short-chain carbohydrates that are not absorbed in the small intestine and are therefore fermented in the colon [15]. This diet reduces gas production that is often responsible for bloating and abdominal pain, but not in all patients. Patients without improvement are considered ‘nonresponders’ and require alternative treatment. However, the FODMAP diet has not been recommended for a long time because it can cause nutritional deficiencies [16,17]. FODMAPs are found in wheat, milk, and dairy products, in some fruits and vegetables, legumes, and others [18]. These foods are also rich in various ingredients, including L-tryptophan (TRP). TRP is the main substrate for serotonin. Both deficiency and excess serotonin can cause the gastrointestinal tract to malfunction. Approximately 90% of TRP is metabolized in the gastrointestinal tract, including along the serotonin pathway. TRP with the participation of tryptophan hydroxylase (TPH-1) is converted to 5-hydroxytryptophan and then to 5-hydroxytraptamine (serotonin) and N-acetyl-5-metoxytryptamine (melatonin; Figure 1). These metabolites have a significant effect on the gastrointestinal tract, as well as on mood [19].

The human diet is composed of various nutrients. These compounds and their metabolites may exert beneficial or unfavorable effects. Many patients report abdominal symptoms after eating, and certain foods ingested can cause exaggerated gastrointestinal complaints. For this reason, the optimal diet for IBS patients is constantly sought. 

The purpose of the current research was to assess the effect of lowering the tryptophan diet on the serotonin pathway and on abdominal and mental symptoms in patients with IBS-D.

## 2. Results

### 2.1. Baseline Comparison between Control and Intervention Group

The general characteristics of the subjects included in the study; the selected laboratory data and the daily tryptophan intake are presented in Table 1.

The exponents in both groups had a similar value, with the exception of the C-reactive protein and fecal calprotectin, which were higher in patients with IBS-D but did not exceed the upper limit of the laboratory standards. Daily tryptophan consumption in the control group ranged from 940 mg to 1629 mg, and in IBS-D patients from 1009 mg to 1728 mg (an insignificant difference). Based on body weight, TRP consumption was lower in the controls than the IBS-D patients (Table 1, *p* > 0.05). Compared to the control group, significant differences were found between the results of the HAM-A score and HAM-D score. Sleep disturbances were more severe in IBS-D patients (Table 1).

Compared to the control group, patients with IBS-D had a higher serum level of serotonin and a lower level of melatonin. The urinary excretion of 5-HIAA was also higher in patients with IBS-D (Table 2).

### 2.2. Pre and Post-Intervention Findings for the IBS-D Group

A positive correlation between the severity of abdominal symptoms (GSRS) and the serum level of serotonin (*p* = 0.02), as well as between the excretion of urinary 5-HIAA (*p* = 0.01), was found, but not between the serum melatonin level and the excretion of urinary aMT6s (*p* = 0.4 and *p* = 0.3, respectively, Table 3).

After nutritional treatment, the serum serotonin levels and 5-HIAA urinary excretion decreased significantly, whereas the serum melatonin levels and urinary excretion of aMT6s did not change significantly. Mental disorders, mainly anxiety, also decreased, but no significant impact on sleep quality was found (Table 4).

A large improvement was found in the somatic state, as shown in Figure 2. Within 12 weeks, diarrhea and abdominal pain, as well as other symptoms, resolved in 33 (82.5%) patients. In the remaining seven (17.5%) patients, pain decreased, but the fluctuations and bloating persisted. The BMI and the results of the routine laboratory tests did not change significantly. The patients also did not report any side effects. 

## 3. Discussion

The results obtained support our [20] and others’ [21,22,23,24] findings of increased serotonin in patients with IBS-D. Most probably, it is linked with the elevated expression of TPH-1 or the decreased activity of SERT [25], as well as an increased availability of free tryptophan as a precursor to serotonin due to an inadequate diet. It has been suggested that free Trp levels may be increased in IBS-D patients. The molecular mechanism underlying the high concentration of TRP is attempted to be related to its reduced oxidation [26]. As a result, TRP alongside four other fecal metabolites, namely, cadaverine, putrescine, phenylalanine, and threonine, has been recently suggested as a disease-relevant potential biomarker of IBS-D [27]. What is noteworthy is that none of the above studies investigated the connection between TRP consumption and serotonin levels. We decided to modify the diet of IBS-D patients to limit the TRP intake by eliminating or limiting products containing a high amount of TRP; serum serotonin levels decreased as a consequence of this. The rationale for the use of a low-TRP diet in patients with IBS-D was significantly higher serotonin secretion compared to healthy people and its positive correlation with the severity of abdominal symptoms. In our study, daily TRP intake in the IBS-D group was relatively high, over 20 mg per kilogram body weight. An average daily intake of TRP is considered to be 3.5–5.0 mg per kg of body weight, and this amount is required In order to maintain the nitrogen balance in the human body. The consumption of TRP in the world is varied and depends on regional dietary habits. For example, in the US population, the mean consumption was found to be 826 mg per day. These levels of TRP intake were safe and were not related to most markers of liver function and kidney function, or to carbohydrate metabolism. Furthermore, such a dose of tryptophan was inversely associated with the level of depression and positively associated with the duration of sleep [28]. The daily consumption of TRP, even at a dose 5.0 g, is safe [29]. Such high doses of tryptophan are used to treat mental illness [30,31], but their use is limited by side effects, such as headaches and dizziness [32]. In turn, insufficient tryptophan intake causes a decrease in serotonin levels, and altered mental mood [33,34], which have also been observed in IBS patients [35]. The people we studied consumed relatively too much TRP, which can cause symptoms of IBS in addition to other factors. After lowering TRP consumption by an average of 50% for three months, it did not cause changes in routine laboratory tests, but serotonin secretion was significantly lower. In addition, abdominal problems, as well as symptoms of mental disorders, resolved or decreased in all patients.

The study was not a randomized controlled trial. The results obtained may inspire further research to determine a balanced diet with an optimal intake of tryptophan, and its impact on the somatic and mental state of patients with IBS. Furthermore, the results obtained did not challenge the accuracy of the FODMAPs diet in the treatment of the IBS but indicate the desirability of modulating TRP intake. This suggestion applies mainly to countries where the consumption of foods rich in this amino acid is high. Moreover, our study provides a potential link between diet and ISB-D symptoms, through altered Trp levels.

## 4. Materials and Methods

### 4.1. Patients

The study included 40 healthy subjects without complaints (Controls) and 40 patients with IBS-D, 23–61 years, recruited in 2017–2021. The control group consisted of healthy people with any GI symptoms, and their laboratory results served as a reference baseline for IBS-D patients. According to the Rome IV criteria, the IBS-D group was characterized by loose or watery stool, which occurred at least 25% of the time for six months. The number of loose stools in the studied patients ranged from 3 to 8 a day and did not coexist with occasional constipation. Furthermore, these patients suffered from abdominal pain related to defecation, bloating, and flatulence. In addition, patients complained of anxiety and sleep disturbances. Only patients for whom the previous FODMAP diet did not resolve their complaints were included in the study.

### 4.2. Diagnostic Procedures

The severity of abdominal symptoms was assessed using the gastrointestinal symptom rating scale—irritable bowel syndrome (GSRS-IBS) [36]. This scale includes 13 gastrointestinal symptoms for the last seven days. The items measure the severity of abdominal pain (1); the pain relieved by bowel action (2); bloating (3); passing gas (4); constipation (5); diarrhea (6); loose stool (7); hard stools (8); the urgent need for bowel movement (9); incomplete intestinal emptying (10); fullness shortly after a meal (11); fullness long after eating (12); and visible distension (13). The items are scored between 1 and 7 points. To determine another GI tract, all patients underwent endoscopic and histological examination of the gastric, duodenal, small intestinal, and colonic mucosa. At the outset, the patients themselves assessed their mental health, and then everyone was assessed for mental health using the Hamilton anxiety rating scale (HAM-A) and the Hamilton depression rating scale (HAM-D). European standards have been adopted for both scales: 10–19 points—mild anxiety/depression, 19–29 points—moderate anxiety/depression, and more than 30 points—severe anxiety/depression. Sleep quality was estimated by the insomnia severity index (ISI) with seven questions and with our own modification, replacing the quality of life (0–4 points) with the shortening of sleeping time. The total score categories were as follows: 0–7 points—no significant insomnia, 8–14 points—subthreshold insomnia, 15–21 points—moderate insomnia, and 22–28 points—severe insomnia.

SIBO was ruled out based on the results of the lactulose hydrogen breath test using a Gastrolyzer (Bedfont, Ltd., Harrietsham, UK). Other exclusion criteria were H-pylori-induced gastritis; lymphocytic and ulcerative colitis; Crohn’s disease; celiac disease; allergy and food intolerance; parasitic and bacterial diseases; liver and renal diseases; diabetes; severe anxiety or depression; and the use of antibiotics, probiotics, and psychotropic drugs in the month before enrollment in the study.

### 4.3. Laboratory Tests

The following routine laboratory tests were performed in all subjects: blood cell count; protein quantification; glucose; glycated hemoglobin; profile of lipids; bilirubin; iron; urea; creatinine; thyroid-stimulating hormone; free thyroxine; free triiodothyronine antibodies against tissue transglutaminase; deaminated gliadin peptide; and the activity of alanine and asparagine aminotransferase, alkaline phosphatase, gamma-glutamyltranspeptidase, amylase, and lipase.

The serum concentration of C-reactive protein (CRP) was determined by a latex agglutination photometric assay in COBAS INTEGRA 800 (Roche Diagnostic, Basel, Switzerland). The fecal calprotectin (FC) was evaluated by a sandwich ELISA test in Quantum Blue Reader (Buhlmann Diagnostics, Amherst, NH, USA). The serum levels of serotonin, as well as the melatonin levels and 5-hydroxyindoleacetioc acid (5-HIAA) and 6-sulafatoxymelatonin(aMT6s) concentrations in urine, were determined by the ELISA method using the Immuno Biochemical Laboratories kit (IBL International GMBH, Hamburg, Germany: No. RE 59121; No. RE 59021; No. RE 59131; No. RE 54031, respectively). Venous blood and 24-h urine collection were centrifuged, and samples were stored at −70 °C. The measurements were performed by photometry at a wavelength of 450 nm (Expert 99 MicroWin 2000 Reader, BMG Labtech, Offenburg, Germany). The concentrations obtained of 5-HIAA and aMT6s in urine were converted from microgram to microgram/24 h.

### 4.4. Dietary Treatment

Nutritional intervention was related only to IBS patients as the control group did not require nutritional intervention. All individuals were recommended to record the type and amount of products consumed every day for 14 days to investigations in the nutritional diary. The average daily TRP intake was calculated using the nutritional calculator with the Kcalmar.pro-Premium application (Hermex, Lublin, Poland). The patients applied a balanced diet of a total caloric value of 2000 kcal with daily intake of a minimum of 50 g of protein, 270 g of carbohydrates, and 70 g of fat. On the day of the evaluation, everyone received a diet with the TRP content calculated in advance. Products with a rich TRP content, including or limited use of wheat bread, sweets, hard cheeses, meat, and cold cuts of some fish, as well as raw fruit and vegetables, were excluded from the diet, but the optimal amount of protein, carbohydrates, and fats was maintained. Subsequently, the test people were instructed to diet 10 mg of TRP per kilogram of body weight per day, for 12 weeks, and it was recommended to complete a diet diary daily, under the control of the nutritionists. After each week, the amount of TRP intake was analyzed to evaluate compliance with the recommendations. Follow-up medical examinations with the evaluation of the somatic and mood symptoms and laboratory tests were performed after 12 weeks.

The study was carried out according to the Declaration of Helsinki and good clinical practice and was approved by the Bioethics Committee (RNN/176/18/KE).

### 4.5. Data Analysis

The normality of the data distribution was checked using the Shapiro–Wilk W test. The U Manna–Whitney test was used to compare differences between two groups. The correlations between the quantitative variables were analyzed using the Spearman rank test. The differences within groups before and after treatment were analyzed using the Wilcoxon signed rank test. All statistical analyses were performed with STATISTICA 13.3 software (TIBCO Software Inc., Palo Alto, CA, USA).

## 5. Conclusions

Lowering the dietary intake of tryptophan may reduce abdominal symptoms and does not disturb the mental state of patients with diarrhea-predominant irritable bowel syndrome.

## Figures and Tables

**Figure 1 ijms-23-15314-f001:**
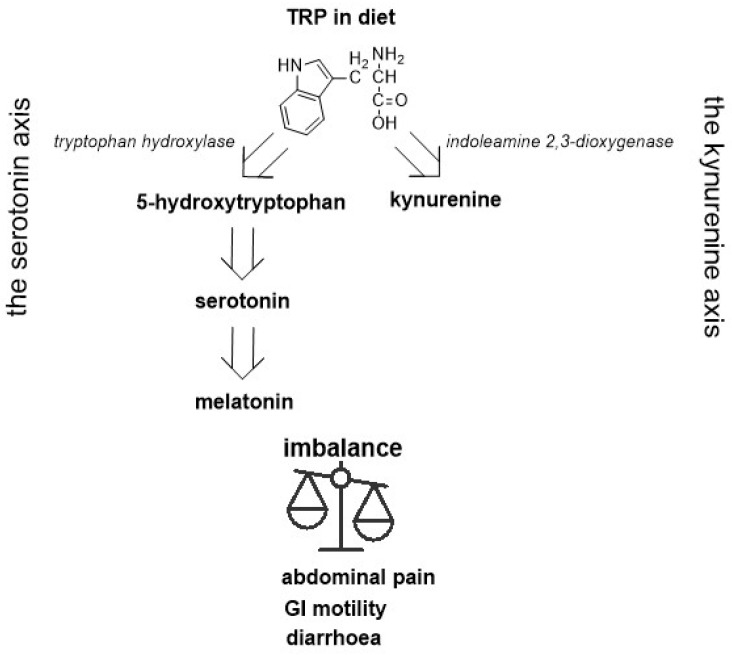
Diagram showing the metabolic pathway involving serotonin in the context of pathophysiology of IBS.

**Figure 2 ijms-23-15314-f002:**
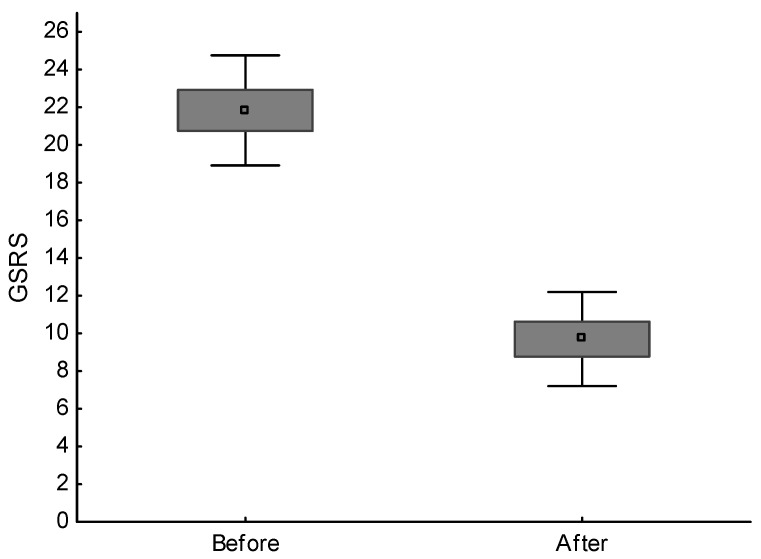
Average and standard deviations of gastrointestinal symptom rating scale (GSRS-points) in patients with diarrhea-predominant irritable bowel syndrome before and after nutritional treatment; difference is significant (*p* < 0.001).

**Table 1 ijms-23-15314-t001:** Characteristics of the healthy subjects (controls, n = 40) and patients with diarrhea predominant irritable bowel syndrome (IBS-D, n = 40) included in the study, and the selected laboratory baseline data, tryptophan intake, and severity symptoms; mean (SD).

Feature ^1^	Controls-Mean (SD)	Patients-Mean (SD)	*p*-Value
Age (years)	44.2 (7.6)	45.9 (9.3)	*p* > 0.05
Gender (M/F)	17/23 (42.5/57.5%)	14/26 (35.0/65.0%)	*p* > 0.05
BMI (kg/m^2^)	23.9 (1.8)	22.6 (2.1)	*p* > 0.05
CRP (mg/L)	1.48 (0.3)	6.5 (2.9)	*p* = 0.038
FC (µg/g)	24.3 (6.2)	45.4 (19.5)	*p* = 0.048
AST (U/L)	16.1 (3.2)	17.3 (9.5)	*p* > 0.05
ALT (U/L)	17.3 (4.8)	22.1 (7.9)	*p* > 0.05
Amylase (U/L)	36.8 (13.1)	43.2 (21.9)	*p* > 0.05
Lipase (U/L)	73.4 (21.3)	48.3 (16.6)	*p* > 0.05
Creatinine (mg/dL)	0.74 (0.2)	0.88 (0.2)	*p* > 0.05
GFR (mL/min)	96.5 (10.7)	89.5 (11.1)	*p* > 0.05
TRP (mg daily)	1269 (191)	1324 (223)	*p* > 0.05
TRP (mg/kg)	21.8 (3.8)	23.2 (4.1)	*p* > 0.05
HAM-A	7.85 (1.2)	21.3 (4.7)	*p* = 0.00002
HAM-D	8.56 (2.6)	14.4 (2.01)	*p* = 0.004
ISI	14.5 (1.6)	9.12 (3)	*p* = 0.00003

^1^ average (SD, standard deviation), BMI—body mass index, CRP—C-reactive protein, FC—fecal calprotectin, AST—aspartate aminotransferase, ALT—alanine aminotransferase, GFR—glomerular filtration rate, and TRP—tryptophan intake.

**Table 2 ijms-23-15314-t002:** Components of the serotonin pathway in healthy subjects (Controls) and in patients with diarrhea—predominant irritable bowel syndrome (IBS-D).

Compound	Controls-Mean (SD)	IBS-D-Mean (SD)	*p*-Value
SER (ng/mL)	142.3 (36.4)	198.2 (38.1)	*p* = 0.0009
MEL (pg/mL)	9.4 (2.4)	8.6 (1.2)	*p* > 0.05
5-HIAA (mg/24 h)	6.0 (1.7)	7.7 (1.5)	*p* = 0.0004
aMT6s (µg/24 h)	11.2 (4.3)	10.6 (2.4)	*p* > 0.05

SER–serotonin, MEL—melatonin, 5-HIAA—5-hydroxyindoleacetic acid, and aMT6s—6-sulfatoxymelatonin.

**Table 3 ijms-23-15314-t003:** Correlation between gastrointestinal symptom rating scale (GSRS) and serum serotonin (SER) and melatonin (MEL) levels, and between GSRS and urinary 5-hydroxyindoleacetic acid (5-HIAA) and 6-sulfatoxymelatinin (aMT6s) excretion, in patients with diarrhea-predominant irritable bowel syndrome.

A Pair of Variable	Spearmans’rho	*p*-Value
GSRS and SER	0.3567	0.0241
GSRS and MEL	−0.1605	0.4012
GSRS and 5-HIAA	0.4216	0.0105
GSRS and aMT6s	−0.2083	0.2718

**Table 4 ijms-23-15314-t004:** Compounds of the serotonin pathway and severity symptoms in patients with IBS-D before and after nutritional treatment.

Feature	Before-Mean (SD)	After-Mean (SD)	*p*-Value
SER (ng/mL)	198.2 (38.1)	123.2 (24.6)	*p* = 0.00002
MEL (pg/mL)	8.6 (1.2)	7.8 (1.9)	*p* > 0.05
5-HIAA (mg/24 h)	7.7 (1.5)	5.5 (0.8)	*p* = 0.00007
aMT6s (µg/24 h)	10.6 (2.4)	10.5 (1.9)	*p* > 0.05
GSRS (points)	21.8 (2.9)	9.7 (2.5)	*p* = 0.00002
HAM-A (points)	19.3 (4.7)	14.9 (3.4)	*p* = 0.00002
HAM-D (points)	14.4 (2.1)	12.9 (3.2)	*p* > 0.05
ISI (points)	14.5 (1.6)	13.8 (1.5)	*p* > 0.05

GSRS—gastrointestinal symptom rating scale, HAM-A—Hamilton anxiety rating scale, HAM-D—Hamilton depression rating scale, and ISI—insomnia severity scale.

## Data Availability

All data are available from corresponding authors on reasonable request.

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
