# Peer review of "A Reduced Tryptophan Diet in Patients with Diarrhoea-Predominant Irritable Bowel Syndrome Improves Their Abdominal Symptoms and Their Quality of Life through Reduction of Serotonin Levels and Its Urinary Metabolites"

_ijms, 2022, doi:10.3390/ijms232315314_

Round 1
Reviewer 1 Report
In this study, the authors investigate the effect of a Tryptophan lowering diet (TURP) on abdominal symptoms and mental health in patients with IBS-D. The authors demonstrate that reduced TRP diet in selected patients with IBS-D results in reduced serum levels of serotonin and its urinary metabolites, with an overall positive impact on the physical symptoms of patients as well as their mental health. The study methodology and findings are well presented. Statistical methods are clear and appropriate statistical analyses are included. It is a well written piece of work with interesting findings, which if replicated in a wider and independent cohort may be useful to patients with IBS-D, widely prevalent in the community.
Whilst the study findings are interesting and well presented, I do have a few questions and comments:
Title- The title of the paper does not accurately reflect the findings. The authors should avoid the use of presumptions in the title such as “through the serotonin pathway”. I propose rewording of the title to something that precisely reflects the study findings such as “A reduced tryptophan diet in patients with diarrhoea-predominant irritable bowel syndrome improves the abdominal symptoms and quality of life through reduction of serum serotonin levels and its urinary metabolites”.
INTRODUCTION
The authors could include a diagram showing the metabolic pathway involving serotonin in the context of pathophysiology of IBS.
METHODS
Line number 222- IgG-dependent food intolerance tests cannot exclude intolerances. There is no robust scientific evidence supporting the use of IgG based food intolerance tests. IgG based tests merely indicate exposure or rather a normal immune response to food allergens. I am aware that these are widely used in the private sector but unfortunately the tests are not scientifically validated. I suggest that the authors acknowledge this in the discussion.
RESULTS
Line 117- needs to be reworded “but sleep quality did not deteriorate”. It could be changed to “but no significant impact on sleep quality”.
Line 129-“somatic state” could be changed to physical symptoms.
DISCUSSION
Lines 139 - 155: This is probably not required in the discussion. The authors could include this in the introduction rather than discussion, avoiding overlap with the contents in the introduction.
The discussion section should probably start with line 168- “the results of our research….
In this line, please avoid the use of the word ‘confirm’. I suggest using ‘indicate’ or ‘demonstrate’. Confirmation would need to be supported by an independent study.
The discussion section may need to be reworded and better presented.
Other comments- the manuscript will need editing for typos and grammatical errors.
Reviewer 2 Report
I commend the authors for looking at different dietary methods for improving IBS symptoms, FGID are a huge healthcare burden. However I have some concerns with the manuscript that need addressing and it is not possible to review this paper completely without this information.
Overall
Many sentences are incomplete or do not make sense – please proof read the manuscript carefully.
It is very unclear throughout what role the control group had other than the initial comparison. You have not made it at all clear whether the controls also had a GI symptom assessment, or whether they also took part in the intervention to assess how they also responded?
Titles of tables and figures need to be improved as little description is provided
Abstract
Use 1 decimal point for results values
Intro
Many sentences are incomplete or do not make sense – please proof read carefully.
‘Proper diet’ cited as references 5-7 needs to be expanded.
The sentence ‘Nevertheless, it has not been recommended for a long time because it can cause nutritional deficiencies and dysbiosis [16,17]’ is very strong based on little recent evidence. Does this really tally with the general literature and clinical practice? Please check this statement more thoroughly.
Sentence ‘in an earlier study’ doesn’t fit well with rest of intro as discusses SIBO, not IBS. Please make more relevant.
There is too little information about Trytophan and the serotonin pathway – there are only 2-3 sentences in the whole introduction but is the basis of the whole paper. Please expand considerably.
Results
Please reduce the number of decimal points throughout the text and tables.
You could add in the HAM-A, HAM-D and ISI scores in to table 1 to simplify this text.
Table 1 should also include GI symptoms too – you present plenty of lab data for comparison that were then not mentioned further. The methods are unclear as to whether the control group also underwent a GSRS-IBS assessment - this should have been done on both groups for comparison as it is unlikely that none of the 40 ‘healthy’ subjects had any GI symptoms?
You jump from presenting controls vs patient data to then using before/after comparisons with no information on whether all subjects did the low tryptophan diet? Are the results the whole group or just IBS?
Discussion
Most of this section should be in the introduction – you are just providing background.
There is very little comparison of the study results with the literature.
Methods
Not sure why this section is after the discussion and not after the introduction? It makes no sense to present results before readers have been able to assess methodological rigour.
It is very unclear here, and throughout the manuscript, whether both the IBS and controls underwent the intervention? This needs to be made VERY clear in every aspect, from abstract, intro, methods, results, discussion. The review of the utility of your findings will be greatly affected by whether the control group provided a true comparison for the intervention, not just the initial baseline data?
Table 1 – the title incorrectly states the participants have SIBO-D. Please add your SD in brackets without the +/- and make clear in the heading that the figures represent: mean (SD). Provide a % for your gender data. Ensure this table heading makes clear it is baseline data.
Figure 2 – Please improve the title of this, please label the axis with ‘before’ and ‘after’ instead of a and b
Figure 3 – Please improve the title of this, there is no need or the red line on this graph, please label the figure with ‘before’ and ‘after’ instead of a and b
Round 2
Reviewer 2 Report
Thank you for making revisions to your manuscript. However, not all my previous comments were addressed and the paper still requires revisions, in particular the discussion which remains an inadequate comparison of your results against the wider literature, or proposed implications of your findings. In addition you have not sufficiently addressed my request to make it very clear throughout that controls are only used as a baseline comparator.
Abstract
You still have not made it clear that the control subjects were only used to compare baseline data – it reads as if they completed the whole study.
Introduction
There are still many incomplete sentences
No need for the TRP diagram
Results
Please use headings for your results as you have not addressed some of my previous comments sufficiently and it remains unclear in this section that controls were baseline only. You still jump from presenting controls vs patient data to then using before/after comparisons with no indication that you are now presenting data for the nutritional intervention for IBS patients only.
A number of my previous comments have not been addressed. Namely, adding a table of GI symptoms for the IBS group – baseline and after intervention.
The results section will be tidied up a lot by adding in the HAM-A, HAM-D and ISI scores in to Table 2 as they would fit well, the title would need revising.
Please add an additional column in to Tables 1, 2 and 4 to present p values instead of using **’s
You did not respond to my previous comment in some results text and table 3 - please reduce the number of decimal points throughout the text and tables.
Some of your results are repeated three times - in text, tables and figures. Each only need presenting in one format, and if table or figure a brief sentence to highlight main findings.
You still have a red line in Figure 4 which is unnecessary
Discussion
This section is still insufficient and needs expanding according to how your results at baseline and after intervention fit in with the wider literature.
Round 3
Reviewer 2 Report
This round of revisions have really improved the manuscript, thank you for taking the time to address my previous comments. A few very minor points to address.
Title
This should be ‘A reduced tryptophan diet…’
Abstract
Please add the word baseline in the following sentence, as indicated: The study included 40 patients with IBS-D, and 40 healthy subjects served as a baseline reference for IBS-D patients, after excluding comorbidities.
Results
Please add the word ‘baseline’ in the following sentence, as indicated: their laboratory results serve as a baseline reference for IBS-D patients.’
Thank you for adding headings – these, however, should be generic and not a results summary. I suggest similar to as follows:
3.1 Baseline comparison between control and intervention group
3.2 Pre and post-intervention findings for the IBS-D group
Many thanks for adding the p values column in to your tables. Please add the actual results, as you have in Table 3. It is not sufficient to add <0.05 or >0.05 (<0.001 is acceptable) as there is a big difference in readers interpretation of trends depending on the p value (p 0.06 is very different to p 0.87 for example)
Discussion
This section is improved but could still be expanded in places – I leave this to the discretion of the editor.
Author Response
Comments (C): This round of revisions have really improved the manuscript, thank you for taking the time to address my previous comments. A few very minor points to address.
Title: This should be ‘A reduced tryptophan diet…’
Abstract: Please add the word baseline in the following sentence, as indicated: The study included 40 patients with IBS-D, and 40 healthy subjects served as a baseline reference for IBS-D patients, after excluding comorbidities.
Results: Please add the word ‘baseline’ in the following sentence, as indicated: their laboratory results serve as a baseline reference for IBS-D patients.’
Thank you for adding headings – these, however, should be generic and not a results summary. I suggest similar to as follows:
3.1 Baseline comparison between control and intervention group
3.2 Pre and post-intervention findings for the IBS-D group
Answer (A): We have done so
A: Many thanks for adding the p values column in to your tables. Please add the actual results, as you have in Table 3. It is not sufficient to add <0.05 or >0.05 (<0.001 is acceptable) as there is a big difference in readers interpretation of trends depending on the p value (p 0.06 is very different to p 0.87 for example)
C: We agreed that the exact p-value matters. The lower the P-value, the lower the error rate and probability of incorrectly rejecting a true null hypothesis. However, in a practical sense, high p-values indicate that the results are not statistically significant. This means that our data do not provide sufficient evidence to conclude that an effect/difference/relationship exists. It is also important to note that comparing a nonsignificant p-value does not indicate that the two variables are equal. It could reflect a small sample size and/or highly variable data, or the null hypothesis is true. Therefore, we stated that comparing nonsignificant p-value is useless and we left p>0.05 as is.
With many thanks for valuable comments
Tomasz Poplawski and Jan Chojnacki